# miRNA-Guided Regulation of Mesenchymal Stem Cells Derived from the Umbilical Cord: Paving the Way for Stem-Cell Based Regeneration and Therapy

**DOI:** 10.3390/ijms24119189

**Published:** 2023-05-24

**Authors:** Arsinoe C. Thomaidou, Maria Goulielmaki, Antonis Tsintarakis, Panagiotis Zoumpourlis, Marialena Toya, Ioannis Christodoulou, Vassilis Zoumpourlis

**Affiliations:** 1Laboratory of Clinical Virology, Medical School, University of Crete, 71500 Heraklion, Greece; arsithomaidu@gmail.com; 2Cancer Immunology and Immunotherapy Center, Cancer Research Center, Saint Savas Cancer Hospital, 11522 Athens, Greece; mgoulielmaki@ciic.gr; 3Biomedical Applications Unit, Institute of Chemical Biology, National Hellenic Research Foundation (NHRF), 11635 Athens, Greece; tsintarakis@hotmail.com (A.T.); stm02506@uoi.gr (P.Z.); mjtoya@gmail.com (M.T.); christodoulou.ioannis@gmail.com (I.C.)

**Keywords:** mesenchymal stem cells, umbilical cord, miRNAs, differentiation, regenerative medicine, cancer

## Abstract

The human body is an abundant source of multipotent cells primed with unique properties that can be exploited in a multitude of applications and interventions. Mesenchymal stem cells (MSCs) represent a heterogenous population of undifferentiated cells programmed to self-renew and, depending on their origin, differentiate into distinct lineages. Alongside their proven ability to transmigrate toward inflammation sites, the secretion of various factors that participate in tissue regeneration and their immunoregulatory function render MSCs attractive candidates for use in the cytotherapy of a wide spectrum of diseases and conditions, as well as in different aspects of regenerative medicine. In particular, MSCs that can be found in fetal, perinatal, or neonatal tissues possess additional capabilities, including predominant proliferation potential, increased responsiveness to environmental stimuli, and hypoimmunogenicity. Since microRNA (miRNA)-guided gene regulation governs multiple cellular functions, miRNAs are increasingly being studied in the context of driving the differentiation process of MSCs. In the present review, we explore the mechanisms of miRNA-directed differentiation of MSCs, with a special focus on umbilical cord-derived mesenchymal stem cells (UCMSCs), and we identify the most relevant miRNAs and miRNA sets and signatures. Overall, we discuss the potent exploitations of miRNA-driven multi-lineage differentiation and regulation of UCMSCs in regenerative and therapeutic protocols against a range of diseases and/or injuries that will achieve a meaningful clinical impact through maximizing treatment success rates, while lacking severe adverse events.

## 1. Introduction

Mesenchymal stem cells (MSCs) are multipotent stromal cells of adult or fetal origin, characterized by the unique abilities of self-renewal, differentiation, and tissue regeneration [1,2], while they have also been shown to induce anti-inflammatory and immunosuppressive responses [3]. Although MSCs can be detected in almost all post-natal tissues, bone marrow (ΒΜ) and adipose tissue (AΤ) are rich and popular sources for MSC isolation [3]. ΒΜ- and AT-originated MSCs share a number of common characteristics, including specific cell surface marker expression, plastic adherence, and the capacity to differentiate into cells of mesenchymal lineage [4]. Apart from their phenotypic resemblance to MSCs of adult origin, fetal MSCs that can be found in fetal, perinatal, or neonatal tissues, such as cord blood (CB), umbilical cord (UC), and placenta [5,6], bear strong advantages [7]. Among these is the higher proliferation capacity and karyotypic stability in cell culture, greater ease of ex vivo genetic manipulation, increased survival during cryopreservation, as well as increased response to environmental stimuli and hypoimmunogenicity [8,9]. Both endogenous and exogenous MSCs have the ability to home to injured sites within the body, where they interact with the local microenvironment to repair the damaged tissue [10,11]. The unique properties of MSCs, especially those derived from the peri- and neonatal tissues, render them prime candidates for regenerative and therapeutic applications. To date, MSCs have been successfully applied to the treatment of the injured myocardium, skin, pancreas, and bone, as well as against several cancer types and other disorders [12,13].

MicroRNAs (miRNAs) are limited-sized RNA sequences with key regulatory properties that differ from protein coding [14]. Primary miRNAs are biosynthesized inside the nucleus via canonical or non-canonical pathways and then transported to the cytoplasm to be processed into precursor and finally mature elements [15]. These active miRNAs can either suppress or infrequently promote gene expression via binding on distinct sites of their target genes, either locally or at distant sites through microvesicle-mediated translocation [16,17]. In detail, the principal role of an activated miRNA is to detect its target; this is accomplished via the recognition of highly conserved complementary sequences in specific sites of the target mRNA. Most often, miRNAs engage to the 3′ untranslated region (3′ UTR) of their target mRNA, thereby inducing mRNA degradation and translational pause. In less common cases, miRNAs may interact with other loci, including the 3′ and 5′ UTRs, gene promoters, and other untranslated sequences to promote their functions [15,18,19]. More specifically, target gene silencing is performed with the complicity of the minimal miRNA-induced silencing complex (miRISC). The miRISC binds to the target mRNA to block its translation, possibly through eIF4F complex interference [20]. Interestingly, it has been proposed that miRNAs can also regulate transcriptional and post-transcriptional mRNA modification and nuclear degradation via a low molecular weight miRISC-guided mechanism that is yet to be elucidated [21]. Since miRNA signaling is an integral part of normal organismic development and maintenance, aberrant expression of specific miRNAs has been linked to certain abnormalities and diseases [22]. Recently, it was shown that miRNAs regulate multiple functions and properties of both embryonic and adult stem cells, including stemness preservation, self-renewal, and differentiation potential [23,24], while MSC-derived exosomes are being exploited as targeted microRNA delivery systems in different diseases [25].

The present review aims at unraveling the mechanisms that govern miRNA-directed differentiation of MSCs, with a special focus on umbilical cord-derived mesenchymal stem cells (UCMSCs). Specifically, UCMSCs’ differentiation potential toward bone, liver, cartilage, neurons, epithelia and other tissues and organs, in the context of miRNA upregulation or suppression, is discussed. Additionally, putative applications of human UCMSCs in conjunction with self-extracellular vesicles carrying specific miRNAs involved in critical cellular functions are described as appealing candidates in the context of stem-cell-based therapies.

## 2. miRNA-Guided Differentiation Potential of UCMSCs

Multipotent stem cells from multiple sources possess a multi-lineage differentiation potential that can be further exploited in several clinical settings, including tissue regeneration and organ rehabilitation for therapeutic and/or aesthetic applications, therapeutic interventions of distinct syndromes and disorders, as well as cancer treatment. When it comes to UCMSCs, this differentiation potential and accordingly the range of applications are significantly enhanced. A summarized report of the miRNAs that are known for driving distinct differentiation fates of UCMSCs is depicted in Figure 1, while the up-to-date recorded research on the role of different miRNAs in the differentiation process of perinatal MSCs is displayed in detail in Table 1.

### 2.1. Osteogenic/Osteoblastic Differentiation

miR-21 appears to play an important role in the osteogenic differentiation of UCMSCs, since human UCMSCs transfected with miR-21 have demonstrated increased expression levels of several known osteogenic genes, such as alkaline phosphatase (ALP), runt-related transcription factor-2 (RUNX-2), and osteocalcin (OCN), an effect that is possibly achieved through the targeting of the PI3K/β-catenin (PI3K-AKT-GSK3β) pathway by miR-21 [31]. Accordingly, genetic or chemical depletion of miR-21 in human UCMSCs (hUCMSCs) resulted in the suppression of their osteogenic properties, via targeting the Wnt/β-catenin pathway [30].

miR-342-3p levels were also found to be significantly upregulated during osteogenesis [28]. Their overexpression in vitro was shown to promote the osteogenic differentiation of hUCMSCs through the elevated expression of various markers, such as OCN and ALP [28], as well as the possible regulation of the transforming growth factor beta (TGF-β) [50] and the sonic hedgehog (Shh) signaling pathways [28], which are generally found to be overactivated during the osteogenic process. However, the findings regarding the role of the Shh signaling in the osteogenic differentiation of mesenchymal stem cells (MSCs) appear to be rather controversial in the literature. A different study demonstrated that the Shh signaling pathway can also act as a suppressor of the osteogenic differentiation of human umbilical cord blood-derived mesenchymal stem cells (hUCBMSCs), possibly via negative regulation of the miR-148 family members, especially miR-148b, which is one of its downstream targets [51]. This could very well explain the pathway’s anti-osteogenic properties in hUCBMSCs, since the expression levels of miR-148b, along with several other microRNAs, including let-7i, miR-23b, miR-141, miR-148a, and miR-152, were found to be significantly upregulated during osteogenic differentiation [51].

miR-196a-5p has been reported to favor the in vitro osteogenic differentiation capacity of Wharton’s jelly-derived mesenchymal stem cells (WJMSCs) by enhancing the activity of several osteogenic markers such as ALP, OCN, Dentin Matrix Protein 1 (DMP1), Bone Sialoprotein (BSP) and Dentin Sialophosphoprotein (DSPP), while also suppressing their proliferative potential. Furthermore, miR-196a-5p overexpression in a calvarial bone defect (segmental defect, SD) rat model was shown to promote defect closure and bone regeneration within 12 weeks after sheet transplantation [29]. These findings appear to be also in agreement with another study that reported that overexpression of miR-196b-5p, a micro-RNA belonging in the same family as miR-196a, results in the inhibition of cell cycle progression and proliferation of WJMSCs [52], potentially favoring a differentiating fate over the preservation of stemness.

It was reported that miR-210 overexpression could potentially lead hUCBMSCs to differentiate into osteoblasts via the upregulation of ALP, OCN, and RUNX-2 during various steps of the differentiation process [53]. miR-216a was found capable of reversing the inhibiting effects of dexamethasone on the osteogenic process, while its expression levels were shown to positively correlate with osteoblastic differentiation and bone formation in adipose tissue-derived mesenchymal stem cells (ATMSCs) and UCMSCs [35]. Furthermore, it was demonstrated that an ex vivo engineered vector combining miR-424 and the bone morphogenetic protein-2 (BMP2) encoding gene has the capacity to promote osteogenesis in WJMSCs, with miR-424 potentially leading to the overexpression of OCN and BMP2 protein levels, the latter of which is considered an important regulator of the osteogenic/osteoblastic differentiation, and ultimately bone development [26].

On the other hand, other microRNAs appear to promote self-renewal and an undifferentiated state over a specific fate determination path. In contrast to miR-424, miR-140-5p has shown the exact opposite effect on BMP2 by suppressing its expression. miR-140-5p and BMP2 endogenous levels are inversely related at distinct time points during the osteogenic process of undifferentiated MSCs derived from multiple sources including the adipose tissue, the bone marrow, and the umbilical cord, while miR-140-5p inhibition seems to allow a significant increase in the expression levels of the osteogenic markers RUNX-2, ALP, OCN, and OPN [27]. Similarly, miR-132 was demonstrated to function as a negative regulator of the osteogenic differentiation of hUCMSCs, via inhibition of the Wnt/β-catenin pathway and suppression of the expression of Osterix, a transcriptional factor required for the osteoblastic differentiation and development [34]. miR-25-3p and miR-33b-5p were also found to inhibit the expression of SMAD5 and Wnt10b, respectively, both of which are important factors for the signaling and promotion of the osteogenic differentiation of mesenchymal stem cells. However, suppression of those two miRNAs by a novel lncRNA, linc02349, was found capable of effectively reversing their effects and allowing the expression of SMAD5 and Wnt10b, thus facilitating the osteogenic differentiation of hUCMSCs [33].

A number of hUCMSC exosome-secreted microRNAs, including miR-429, miR-34b-3p, miR-370-5p, miR-1270, miR-4454, miR-619-5p, miR-150-5p, miR-365b-3p, miR-365a-3p, miR-10a-5p, miR-615-3p, miR-328-3p, let-7d-3p, miR-675-5p, miR-10b-5p, miR-574-3p, miR-204-5p, miR-433-3p, miR-2110, miR-382-5p, miR-25-3p, miR-345-5p, miR-146a-5p, miR-629-5p, miR-590-3p, miR-21-5p, miR-377-5p, miR-1246, miR-188-5p, miR-329-3p, miR-3074-5p, miR-136-3p, miR-27a-3p, miR-598-3p, miR-30d-5p, miR-299-3p, miR-337-5p, miR-549a-5p, miR-655-3p, miR-410-5p, and miR-4423-5p, were also identified as critical modulators of the osteoblastic proliferation and differentiation processes, with miR-2110 and miR-328-3p highlighted as the most probable of regulating bone differentiation, based on their specific target genes [32]. These 41 miRNAs were also reported capable of reversing ovariectomy-induced osteoporosis to a certain degree in an in vivo mouse model, suggesting that exosome-derived microRNAs might also present great potential for therapeutic application in the clinical setting.

### 2.2. Hepatic Differentiation

The second most-studied field in the umbilical cord mesenchymal stem cell fate determination involving the mediation of microRNAs appears to be the path toward hepatic differentiation. A next-generation sequencing analysis identified a total of 63 miRNAs, both novel and known, that presented significant changes in their expression patterns during different time points of hepatic differentiation of WJMSCs, while a bioinformatics analysis of their target genes shed some light on the molecular pathways and liver-specific transcription factors involved in the hepatic differentiation process [54]. Another miRNA-level analysis of human umbilical cord Wharton’s jelly revealed various significant microRNAs during hepatic differentiation, including the miR-23b cluster (miR-27b-3p, miR-24-1-5p, and miR-23b-3p), miR-26a-5p, miR-30a-5p, miR-122-5p, miR-148a-3p, and miR-192-5p), all of which are suggested to promote stem cell induction into hepatocytes through the negative regulation of targets related to the inhibition of hepatic differentiation, with miR-122-5p specifically suppressing the mesenchymal markers Sex Determining Region-Box 11 (SOX11) and Vimentin (VIM) [36]. A number of 61 unique microRNA profiles were reported as differentially expressed consistently during the hepatic differentiation process, thus promoting or suppressing the differentiation of umbilical cord lining-derived mesenchymal stem cells, accordingly [55]. Seven of those unique miRNAs, including miR-1246, miR-1290, miR-148a, miR-30a, miR-424, miR-542-5p, and miR-122, many of which are normally found upregulated during hepatic differentiation, have been reported capable of stimulating the conversion of UCMSCs into fully functional hepatocytes when concomitantly overexpressed in vitro. Both hepatocytes and MSCs expressing those specific seven microRNAs possessed the ability to repair liver injury and to improve liver function in a CCL4-injured mouse model after two weeks of transplantation [56]. In addition, a more recent study concluded that an optimized combination of only five of those microRNAs, specifically, miR-122, miR-148a, miR-424, miR-542-5p, and miR-1246, could produce the same results in both an in vitro and an in vivo environment, without the need for the presence of miR-30a and miR-1290 [57]. Similarly, the concurrent overexpression of another three-miRNA set, including miR-106a, miR-574-3p, and miR-451, was reported capable of promoting the differentiation of hUCMSCs into mature and functional hepatocytes in vitro [37], suggesting that specific combinations of miRNAs could provide an alternative procedure through which mature hepatocyte-like cells are generated for therapeutic purposes in the future.

### 2.3. Neural Differentiation

A number of miRNAs are also suggested to play an integral part in the neural differentiation of UCMSCs. Several studies demonstrated that there exist statistically important differences in the expression levels of certain microRNAs between WJMSCs trans-differentiated into neuronal cells and undifferentiated WJMSCs. More specifically, as part of a microarray analysis, a total of 161 differentially expressed miRNAs were detected, 28 of which demonstrated a high fold change of more than 5, with miR-4521, miR-222-5p, miR-92a-1-5p, miR-543, and miR-548a-3p presenting significantly downregulated levels and miR-575, miR-4440, miR-297, miR-4793-3p, miR-371b-5p, miR-3617-5p, miR-125b-2-3p, miR-26b, miR-124a, miR-7152-3p, miR-1290, miR-5093, miR-663b, miR-6861-5p, miR-212-5p, miR-194, miR-129-5p, miR-132-5p, miR-1202, miR-3687, miR-195-3p, miR-192-5p, and miR-5572 showing significant upregulation in WJMSC-derived neurons compared to the undifferentiated controls [58]. Similarly, miR-345, miR-106a, miR-17-5p, miR-20a, and miR-20b were characterized as upregulated in undifferentiated WJMSCs, whereas miR-206, miR-34a, miR-374, miR-424, miR-100, miR-101, miR-323, miR-368, miR-137, miR-138, and miR-377 were found to be significantly upregulated in trans-differentiated WJMSCs. Moreover, miR-20a, miR-20b, miR-17-5p, and miR-106a, which all belong to the miR-17 family, were found to be downregulated during the neurogenesis of WJMSCs. Among these, miR-34a overexpression was shown to significantly limit stem cell motility, indicating that the suppression of endogenous miR-34a could possibly be utilized as a method of enhancing transplanted stem cell motility toward injury sites in stem cell-based therapies [59]. Similarly, miR-203 was found to suppress neural retina differentiation through targeting three retina development-related genes, namely, DKK1, CRX, and NRL. However, anti-miR-203 transfection of hUCBMSCs was shown to successfully reverse this inhibiting effect and allow the expression of those target genes, thus converting cells into photoreceptor cell types [39]. The combination of two more microRNAs, miR-20b and miR-106a, might also have an important role to play in the negative regulation of neural differentiation of hUCMSCs, possibly via the suppression of Neurogenin-2 (Ngn2), which is known for controlling the cell cycle progression and promoting neural differentiation [38]. These findings further indicate that stem cells treated with either specific miRNAs or corresponding miRNA inhibitors might provide a reliable source of desired cell types or even subtypes for stem cell-based therapies of non-regenerative diseases.

Another large-scale analysis determined that the expression levels of various miRNAs related to motor neuronal cell differentiation and proliferation are altered in relation to different stages of hUCBMSC differentiation. miR-9-5p and miR-324-5p (related to differentiation) demonstrated significant upregulation during early time points of neuronal differentiation, while miR-137 and miR-let-7b (related to proliferation) showed significant downregulation during the latter stages of differentiation. Different expression profiles were also detected between hUCBMSCs treated with neurogenic induction agents (retinoic acid and Shh in combination), with miR-449c-5p, miR-1249-3p, miR-9-5p, and miR-324 showing elevated expression levels and miR-335-3p and miR-335-5p demonstrating decreased expression in treated cells compared to the untreated controls. Furthermore, this comparison of miRNA profiles between undifferentiated and treated/induced hUCBMSCs revealed a number of new differentially expressed miRNAs, including novel-miR-17, novel-miR-18, and novel-miR-20, which appeared to be significantly upregulated, and novel-miR-1 and novel-miR-2, which showed significant downregulation [60]. It should be noted here that a number of those previously unknown miRNAs were found to target genes in six critical pathways involved in the neuron differentiation process, including the cholinergic synapse, axon guidance, hedgehog, MAPK, TGF-β, and JAK-STAT signaling pathways, although their exact mechanistic functions remain a subject of future investigation [60].

### 2.4. Chondrogenic Differentiation

In regard to chondrogenic differentiation, miR-29b-3p expression was shown to decrease during the chondrogenic process, through targeting of SRY-related high-mobility-group box 9 (SOX9), a crucial transcription factor for chondrocyte differentiation, while, similar to other long non-coding RNAs, lncRNA H19 was observed to function as a sponge of miR-29b-3p, essentially inhibiting its effect and allowing for the expression of SOX9 and the subsequent platelet lysate (PL)-induced chondrogenic differentiation of hUCMSCs [40]. Furthermore, miR-340-5p, miR-130a-3p, miR-381-3p, miR576-5p, and miR-337-3p were found to be highly upregulated in small extracellular vesicles (sEVs) obtained from hUCMSCs treated with Kartogenin (KGN) compared to untreated control cells, whereas miR-200b-3p, miR-200c-3p, miR-375, miR-122-5p, and miR-182-5p appeared to be downregulated in treated compared with untreated stem cells. Among these, miR-381-3p overexpression was shown to be capable of promoting chondrogenic differentiation of MSCs and upregulation of chondrogenesis-related genes, such as SOX9, aggrecan, and collagen II, while directly binding and inhibiting the expression of TAOK1, an upstream regulator of the Hippo signaling pathway [41]. It should be noted that transplanted KGN-sEVs, along with hUCMSCs, were also observed to restore tissue injury in a knee articular cartilage defect rabbit model four weeks after the operation [41], suggesting that sEVs might also have a role to play in the in vivo induction of chondrogenesis as a future therapeutic approach.

### 2.5. Epithelial Differentiation

In terms of MSC epithelial differentiation, miR-145 was shown to promote the conversion of UCMSCs into type II alveolar epithelial cells under hypoxic conditions, possibly via targeting the TGF-β receptor II (TGFβRII), which in turn results in the suppression of differentiation into fibroblasts by the TGF-β signaling pathway [43]. Since the TGF-β pathway is highly upregulated during fibrosis after acute lung injury (ALI), hypoxia-induced miR-145 could present a new method of stem cell-based therapies for lung injury [43]. A number of microRNAs, including miR-100, miR-127-3p, miR-136, miR-146a, miR-199a-5p, miR-214, miR-224, miR-299-5p, miR-337-5p, miR-34a, miR-376a, miR-376c, miR-377, miR-379, miR-381, miR-409-3p, miR-410, miR-424, miR-654-3p, miR-758, and miR-762, which are normally depleted in the retinal pigment epithelium (RPE) and in the ARPE-19 cell line, were found to be overexpressed in undifferentiated hUCBMSCs [42]. Among these, miR-410 was demonstrated to target more than one RPE-related gene, specifically, OTX2 and RPE65, while its inhibition could induce RPE differentiation in human amniotic epithelial stem cells [61]. In agreement with these results, it was later demonstrated that multiple transfections with anti-miR-410 in UCBMSCS could indeed induce upregulation of RPE-specific factors, such as Melanocyte Inducing Transcription Factor (MITF), LRAT, RPE65, Bestrophin, and EMMPRIN, and therefore result in direct RPE differentiation [42].

### 2.6. Differentiation toward Other Lineages

In regard to differentiation into insulin-producing cells (IPCs), miR-200b-3p demonstrated the capability of directing UCMSCs toward this fate via the inhibition of the ZEB2 transcription factor, while its suppression was shown to induce hypoglycemia and to inhibit insulinogenesis in successfully differentiated IPCs from UCMSCs in an in vivo diabetic mouse model [44]. In addition, miR-375 and miR-26a were shown to be highly enriched in undifferentiated chicken nestin-positive UCMSCs (N-UCMSCs) compared with nestin-positive pancreatic mesenchymal stem cells (N-PMSCs). Combined miR-375 and miR-26a transfection was found capable of suppressing several target genes, including mtpn, sox6, bhlhe22, and ccnd, thus inducing IPC differentiation 12 days after treatment. Furthermore, the two microRNAs’ function was further validated in vivo by transplanting successfully differentiated N-UCMSCs in a hyperglycemic mouse model and confirming that IPCs from N-UCMSCs do indeed secrete chicken insulin into the host animals’ blood within two weeks after they are injected with glucose, suggesting that cell-based transplantation therapies for diabetes might become a promising strategy in the future [45].

In regard to myoblastic differentiation, a group of four microRNAs, specifically, miR-21, miR-23a, miR-125b, and miR-145, were shown to be highly upregulated in exosomes derived from UCMSCs. High-throughput RNA analysis revealed that these specific miRNAs have an essential role in the inhibition of myofibroblast and scar formation in both an in vivo skin wound mouse model and in vitro, by suppressing genes in the TGF-β/SMAD2 pathway, such as TGF-β2, TGF-βR2, and SMAD2, as well as α-smooth muscle actin (α-SMA) and collagen I expression, suggesting that UCMSC-derived exosomes could pose an alternative to cell-based therapies to prevent tissue fibrosis and wound scarring in the clinical setting [48]. Additionally, miR-503-5p and miR-222-5p were both identified as contrasting regulators of smooth muscle cell differentiation from hUCMSCs, with miR-503 promoting the differentiation process by directly targeting SMAD7 and miR-222-5p inhibiting the same process through dual targeting of ROCK2 and αSMA [49].

In regard to hematopoietic differentiation, miR-218, miR-150, and miR-451 were all found to be upregulated during the hematopoietic induction of UCMSCs via combined Aza/GF treatment. Among these microRNAs, miR-218 was revealed to negatively regulate the transcription of MITF, while its overexpression might possibly result in the promotion of hematopoietic differentiation via the upregulation of NF-Ya and HoxB4, both of which are important transcriptional inducers of hematopoiesis [46].

In regard to adipogenic differentiation, it was demonstrated that although UCMSCs do possess the capacity to differentiate into adipocytes, their potential does not seem to be as strong compared to that of MSCs derived from different tissue origin, such as adipose tissue or bone marrow, which is an important parameter to consider when it comes to selecting specific stem cell sources for therapeutic purposes [47]. Nevertheless, the miR-301b-miR-130b cluster, which is theorized to inhibit PPARγ, a critical adipogenetic transcription factor, was proposed as a negative regulator of adipogenic induction in MSCs derived from all three sources, with the two miRNAs’ endogenous expression levels directly correlating with each MSC type’s adipogenic potential [47].

Although in most cases the specific molecular mechanisms through which different miRNAs control and regulate the determination of cell fate during the differentiation of hUCMSCs still remain poorly understood, taken together, the above findings do indicate that the overexpression or inhibition of unique or multiple miRNAs could eventually serve as a promising method of efficiently generating functional cells of desired lineages in the clinical setting, especially when it comes to matters of regenerative medicine, as will be discussed further in the following section of this study.

## 3. Applications of UCMSCs in Regenerative and Therapeutic Medicine and the Role of miRNAs

In recent years, especially in the last decades, the potential therapeutic applications of hUCMSCs have been intensively explored for the treatment of various pathologies, as can be deduced from the literature. Human UCMSCs, as well as their secreted exosomes and other extracellular vesicles, which often carry specific miRNAs involved in key cellular functions, appear to possess significant regenerative, anti-inflammatory, protective, and in some cases, even tumor-suppressing capabilities [62]. Their ameliorative properties, along with their strong self-renewal rate, low immunogenic potential, and non-invasive isolation does indeed make UC-derived MSCs one of the most appealing candidate sources for novel stem-cell-based therapies as well as other applications, including general and tissue-specific toxicity screening [63]. Promising applications of UCMSCs in the context of miRNA-directed regenerative/healing and therapeutic potential are presented in Table 2 and Table 3 and in the corresponding Figure 2 and Figure 3, respectively.

### 3.1. Ischemic/Reperfusion (I/R) Injuries

Ischemic-reperfusion injury (IRI) is a critical condition that can affect various organ systems and for which there is currently no effective therapeutic strategy, but mostly supportive treatment. IRI is currently a major cause for transplantation complications and graft loss; however, MSC-based therapies appear to present a promising new method, since a number of different microRNAs, specifically delivered through UCMSC-exosomes, have been reported to play important functional and regulatory roles in the complex biological processes associated with IRI. UCMSC microvesicle-transferred miR-21, one of the most universally studied microRNAs and known for its involvement in the processes of angiogenesis, apoptosis, and inflammation, was found to be capable of ameliorating renal IRI both in vitro and in an in vivo rat model, possibly by regulating programmed cell death protein 4 (PDCD4) and inhibiting tubular epithelial cell apoptosis under hypoxic conditions [67]. Similarly, miR-1246, delivered through UCBMSC-exosomes, appears to pose a novel therapeutic option for hepatic IRI, since its presence was shown to shield hepatocytes against the apoptotic effects of hypoxia/reoxygenation (H/R) damage via decreasing the expression of the pro-inflammatory factors TNF-α, IL-6, IL-1β, and IL-17 [126,127] and regulating the GSK3β/Wnt/β-catenin signaling pathway [126]. miR-1246 was also shown to be capable of restoring the Th17/Treg cell imbalance caused by IRI in the liver, through the IL-6-gp130-STAT3 signaling pathway [127], while I/R rat treatment with miR-20a-containing exosomes appears to mediate the autophagic and apoptotic activity of hepatocytes by inhibiting the expression of Beclin-I, fatty acid synthase (FAS), and active Caspase-3 [66].

miR-24 is another microRNA found to be involved in the protective role that UCMSC-extracellular vesicles (EVs) appear to play in cerebral IRI. miR-24, carried by UCMSC-EVs, was reported to protect the brain tissue from I/R injury and ameliorate the cerebral damage both in cultured cells and in an in vivo rat model via inhibiting the expression of AQP4 and activating the P38 MAPK/ERK1/2/PI3K/AKT pathway [79]. miR-26b-5p was shown to play an important role in the prevention of nerve damage caused by cerebral I/R, since its delivery via UCMSC-exosomes represses microglia M1 polarization and neuroinflammation by negatively regulating inflammatory factor CH25H and inhibiting the activity of the toll-like receptor (TLR) pathway [81]. Additionally, exosome-derived miR-26b-5p was reported to have a protective effect over PC12 neurons in the brain tissue by targeting the neuro-inflammatory factor MAT2A and inhibiting the MAPK/STAT3 pathway, thus allowing for the possible recovery of early brain injury damage [82]. miR-410, delivered via UCMSC-EVs to the neurons of hypoxia-ischemia brain damage (HIBD) neonatal mice, improves the viability of neuronal cells while reducing the apoptotic level of damaged cells by downregulating HDAC1, thus enabling the expression of neuroprotective factors EGR2 and Bcl2 [117]. Furthermore, modified UCMSC-exosomes that underwent knockdown of miR-206, a microRNA that directly targets the neuroprotective BDNF gene, presented significant ameliorating effects in the improvement of impaired brain function and edema, as well as the prevention of neuronal apoptosis [112]. In the case of H/R-induced cardiac injury, UCMSC-exosomes carrying lncRNA UCA1 were suggested to shield cardiac microvascular endothelial cells from I/R damage, with lncRNA UCA1 functioning as a negative regulator of miR-143, which naturally aggravates the effects of oxidative stress in cardiomyocytes [103]. In addition, miR-21, was shown to enhance the production of new blood vessels in critical limb ischemia (CLI) by targeting the carboxyl terminus of Hsc70-interacting protein (CHIP) and increasing the angiogenetic activity of HIF-1a [68].

Transplantation of UCMSC-exosomes, specifically enhanced with silk fibroin hydrogel and carrying miR-675, was also proposed as a novel strategy for the treatment of vascular disease, with miR-675 significantly inhibiting the aging-related vascular dysfunction and promoting blood perfusion in an ischemic hindlimb mouse model [124]. Similar results were reported in the case of skeletal muscle ischemic injury, since combined treatment with WJMSCs and miR-29a has shown a particularly efficient therapeutic potential by greatly enhancing the angiogenetic potential of vein endothelial cells and reversing the impaired blood perfusion in the hindlimb of BPVC-injured mice [85]. Treatment of co-cultured UCMSCs and endothelial colony-forming cells (ECFCs) with hyaluronic acid was shown to cause significant downregulation of miR-139-5p through CD44 activation in order to facilitate the blood flow in the tissues of ischemic hindlimb rats as well as to enhance their angiogenetic properties [100]. The therapeutic properties of UCMSC-exosomes in injured hindlimb mice were also attributed to miR-24, which targets the pro-apoptotic factor Bim, thus enhancing muscle motility and blood flow [80], while recently it was suggested that the exosomal circular RNA circHIPK3 could act as a sponge for miR-421, essentially allowing for the expression of its target, FOXO3a, and the inhibition of pyroptosis and inflammation observed in ischemic mice [118].

### 3.2. Acute Organ Injuries

Acute kidney injury (AKI) is yet another severe clinical syndrome for which there appear to be a rising number of proposed cell-based studies involving UCMSCs. UCMSCs were found to enhance the autophagic potential of HK-2 cells in vitro through the secretion of miR-145, which in turn negatively regulates the PI3K/AKT/mTOR pathway [104]. Treatment with UCMSCs demonstrated a significant increase in the survivability of mice with sepsis-associated AKI through the upregulation of miR-146b, which subsequently blocks NF-κB pro-inflammatory activity via the suppression of its activator protein, IRAK1 [107]. Another recent study demonstrated that not only do UCMSC-exosomes specifically prefer to home to the proximal tubules of injured kidneys, but they also effectively mediate the typical effects of I/R in an AKI mouse model. Via miR-125b-5p delivery, injected exosomes appear to suppress the activation of p53 and protect damaged tubular epithelial cells from G2/M cycle arrest and apoptosis, while further promoting their proliferation and repair [96].

In addition to AKI, UCMSCs possess similar beneficial and regenerative effects over a number of various models of ischemic injury, including acute liver injury (ALI), acute myocardial infarction/ischemia (AMI), and acute spinal cord injury (SCI). UCMSC-exosomes, enriched with miR-455-3p, inhibited the activation of monocytes/macrophages in the liver by targeting PIK3r1 and downregulated the expression of key inflammatory cytokines, such as IL-6, in an in vivo mouse model, thus improving liver response and repair [120]. Exosome-derived miR-451 was reported to alleviate inflammation in burn-induced ALI rats, by significantly downregulating the expression levels of various known inflammatory cytokines, including TNF-α, IL-1β, and IL-6, via the suppression of the TLR4/NF-κB pathway [130], as well as the regulation of the MIF/PI3K/Akt signaling pathway to induce M1 to M2 macrophage polarization [119]. UCBMSC-derived exosomal miR-22-3p also demonstrated therapeutic properties in the prevention of lipopolysaccharide (LPS)-induced ALI. Its delivery reduces inflammatory and oxidative stress responses both in vitro in lung cells and in vivo in rat lung tissues via silencing of inflammatory factors TNF-α, IL-1β, IL-6, and the frizzled class receptor 6 (FZD6) gene [75]. Exosomal miR-377-3p was demonstrated to play a role in the mediation of inflammation in LPS-induced ALI mice by inducing cell autophagy through downregulation of its target gene RPTOR [120], while miR-100 delivery through WJMSC-microvesicles (MVs) appears to function similarly by negatively regulating mTOR expression and increasing autophagy levels in BLM-induced ALI rats [91]. Exosomal miR-100-5p presents significant protective properties over cardiomyocytes that have undergone H/R damage by suppressing the transcription factors FOXO3 and NLRP and consequently shielding cells from the effects of inflammasome activation, cytokine release, and pyroptosis [92].

In like manner, miR-19a in UCMSC-exosomes mitigates the effects of hypoxic damage via the suppression of SOX6 and the regulation of the mitochondrial apoptotic pathway AKT/JNK3/caspase-3 [65]. EV-delivered miR-223 provides yet another potential therapeutic tool for AMI as it both represses the inflammatory response in cardiomyocytes and induces the angiogenic properties of vein endothelial cells, protecting rats from myocardial fibrosis while concurrently promoting myocardial healing via the modulation of the P53/S100A9 axis [113]. A somewhat different approach was provided in the case of miR-125b-5p, as it was suggested that its upregulation might contribute to the effects of myocardium infraction after hypoxic injury. However, UCMSC-exosomes were shown to negate its harmful properties by upregulating Smad7 to suppress its expression levels and facilitate cardiomyocyte repair [95]. Interestingly, migration inhibitory factor (MIF) engineered UCMSC-exosomes demonstrated even more efficient cardioprotective properties in rats in comparison to non-modified UCMSC-exosomes, possibly through the significant upregulation of miR-133a-3p, which in turn activates the AKT signaling pathway and increases VEGF expression to promote ischemic damage repair [98]. UCBMSC-exosomes loaded with miR-23a-3p were shown to significantly decrease H/R-induced myocardial cell ferroptosis in mice via the targeting of the DMT1 gene [77], while the delivery of exosomal miR-136 could potentially revive aged and senescent BMMSCs to improve their cardiac repair functions by inhibiting Apaf1 expression in MI mice [99].

With regards to SCI, miR-29b-3p delivery through UCMSC-EVs was shown to improve the motility and nerve function repair of model rats by downregulating the expression of PTEN and activating the Akt/mTOR pathway [87], while exosomal miR-29b-3p also prevents the apoptotic effects of LPS-induced SCI in damaged neurons via inhibition of PTEN and induction of the PI3K/AKT pathway [88]. Exosomal miR-199a-3p and miR-145-5p function synergistically to promote neuronal cell differentiation and improve the impaired hindlimb motility of SCI rats while decreasing inflammation severity at the spinal cord damage site by suppressing the expression of the Cblb and Cbl genes and subsequently regulating the NGF/TrkA pathway [131]. Similar results were reported following the treatment of SCI rats with miR-126-loaded UCMSC-exosomes, as this particular microRNA appears to reduce apoptosis and enhance both angiogenesis and neurogenesis in the injured spinal cord by potentially inhibiting the expression of its two target genes, SPRED1 and PIK3R2 [97]. Lastly, it was reported that treatment with miR-146a-5p-transfected exosomes leads to the reduction of astrocyte neurotoxicity in the spinal cord through targeting the Traf 6/Irak1/NFκB pathway [106].

**Table 3 ijms-24-09189-t003:** miRNA-guided therapeutic applications of perinatal and neonatal MSCs.

miRNA	Tissue Origin	Vehicle Type	Target (Gene/Pathway)	Function	Clinical Application	Reference
miR-17-3p	UCMSCs	Exosomes	STAT1	Inflammation/Apoptosis Suppression, Oxidative Injury Reduction	Diabetic Retinopathy	[132]
miR-17-5p	UCMSCs	Exosomes	SIRT7	ROS Reduction, Proliferation Promotion	Premature Ovarian Insufficiency	[133]
miR-18b	UCMSCs	EVs	MAP3K1/NF-κB/p65	Apoptosis/Inflammation Inhibition	Diabetic Retinopathy	[134]
	UCMSCs	EVs	Notch2/TIM3/mTORC1	Proliferation/Migration Promotion, Blood Pressure Reduction	Pre-Eclampsia	[135]
miR-21	UCMSCs	EVs	TGF-β2	Myoblast Differentiation Inhibition	Lung Fibrosis	[136]
	UCMSCs	Exosomes	LATS1	Estrogen Secretion Promotion	Premature Ovarian Insufficiency	[137]
	UCMSCs	Exosomes	p38 MAPK	Apoptosis/ER Stress Suppression	Diabetes	[138]
miR-23	UCMSCs	EVs	TGF-βR2	Myoblast Differentiation Inhibition	Lung Fibrosis	[136]
miR-24-3p	UCMSCs	Exosomes	Keap-1	Lipid Accumulation/ROS Generation/Inflammation Inhibition	Non-Alcoholic Fatty Liver Disease	[139]
miR-26a-5p	UCMSCs	Exosomes	METTL14/NLRP3	Cell Survival Promotion, Pyroptosis Inhibition	Intervertebral Disc Degeneration	[140]
miR-27b	UCMSCs	Exosomes	HOXC6	EMT Suppression	Subretinal Fibrosis	[141]
miR-29a	UCMSCs	EVs	HBP1/Wnt/β-catenin	Proliferation Promotion, Apoptosis Inhibition, Ovarian Function Restoration	Premature Ovarian Insufficiency	[142]
	Placenta-derived MSCs	Exosomes	-	Differentiation Promotion, Utrophin Increase, Fibrosis/Inflammation Inhibition	Duchenne Muscular Dystrophy	[143]
miR-30c-5p	UCMSCs	EVs	PLCG1/PKC/NF-κB	Inflammation Suppression	Diabetic Retinopathy	[144]
miR-100	UCMSCs	EVs	HS3ST2	Proliferation/Invasion/Migration/EMT Promotion	Endometriosis	[145]
miR-100-5p	UCMSCs	-	NOX4/NLRP3, GSDMD	Inflammation/Oxidative Stress/Apoptosis Inhibition	Premature Ovarian Insufficiency	[146]
	UCMSCs	EVs	-	M2 Polarization Promotion, Treg Generation	SS Dry Eye	[147]
	UCMSCs	EVs	NOX4	ROS/Oxidative Stress/Apoptosis Inhibition	Heart Failure	[148]
	UCMSCs	Exosomes	FZD5/Wnt/β-catenin	Migration Inhibition, Apoptosis Promotion, Inflammation Suppression	Atherosclerosis	[149]
miR-101	UCMSCs	EVs	BRD4/NF-κB/CXCL11	Proliferation/Migration Promotion	Pre-Eclampsia	[150]
miR-125b	UCMSCs	EVs	IL-6R, IFV genes	Viral Activities/Infection Inhibition	Respiratory Virus-associated Diseases	[151]
miR-126	UCMSCs	Exosomes	HMGB1	Inflammation Suppression	Retinal Inflammation	[152]
miR-126-3p	UCMSCs	Exosomes	PIK3R2, PI3K/AKT/mTOR	Proliferation/Angiogenesis Promotion, Apoptosis Suppression	Premature Ovarian Insufficiency	[153]
miR-133	UCMSCs	Exosomes	-	Proliferation/Survival Promotion, Bregs Production	Immune Thrombocytopenia	[154]
miR-133b	UCMSCs	Exosomes	SGK1	Proliferation/Cell Cycle Progression/Migration/Invasion Promotion, Apoptosis Inhibition	Pre-Eclampsia	[155]
miR-140-5p	UCMSCs	Exosomes	FSTL3	Cell Growth/Angiogenesis Promotion, Inflammation Suppression	Pre-Eclampsia	[156]
miR-146a	UCBMSCs	-	-	Inflammation Suppression	Inflammatory Diseases	[157]
	UCMSCs	Exosomes	SUMO1/β-catenin	Colitis Deterioration/CAC Progression Inhibition	Colitis	[158]
	UCMSCs	Exosomes	TRAF6, IRAK1, NF-κB	Fibroblast Activation/Inflammation/Fibrosis Inhibition	Urethral Stricture Diseases	[159]
	WJMSCs	Exosomes	-	M2 Macrophage Polarization Promotion, Inflammation Inhibition	Inflammatory Disorders/Sepsis	[160]
miR-146a-5p	UCMSCs	Exosomes	TRAF6	Neuroinflammation/Pyroptosis Suppression, Autophagy Promotion	Inflammatory Pain	[161]
	UCMSCs	Exosomes	NOTCH1	Bleeding/Inflammation/M1 Polarization Suppression, M2 Polarization Promotion	SLE-associated DAH	[162]
	UCMSCs	-	TRAF6/STAT1	M2 Polarization Promotion, Renal Function Improvement, Inflammation Suppression	Diabetic Nephropathy	[163]
	UCMSCs	EVs	(TGF-β1/Smad2/3)	Allergic Inflammation/Fibrosis/Airway Remodeling Suppression	Asthma	[164]
miR-147	WJMSCs	EVs	-	Inflammation Suppression, Macrophage Activation	Abdominal Aortic Aneurysm	[165]
miR-148a-5p	UCMSCs	-	Notch2	Proliferation Promotion, Apoptosis/Fibrosis Inhibition	Liver Fibrosis	[166]
miR-153-3p	UCMSCs	-	Snai1	EMT Suppression	Peritoneal Fibrosis	[167]
	UCMSCs	-	PELI1	Proliferation/Migration Inhibition, Tfh/Treg Imbalance Promotion	SLE	[168]
miR-181a	UCMSCs	-	-	T Lymphocyte Regulation	SLE	[169]
miR-195	UCMSCs	Exosomes	TFPI2	Hypoxic Damage Reduction	Pre-Eclampsia	[170]
miR-199	UCMSCs	-	KGF	(Fibrosis Promotion)	Cirrhosis	[171]
miR-199a-5p	UCMSCs	-	Sirt1/p53	CD4+ T-cell Senescence Promotion	SLE	[172]
miR-203a-3p.2	UCMSCs	Exosomes	casp11/4	Macrophage Pyroptosis/Inflammation Inhibition	IBD	[173]
miR-204	WJMSCs, BMMSCs	Exosomes	STAT3	Proliferation Inhibition	Pulmonary Hypertension	[174]
anti-miR-206	UCMSCs	-	BDNF, (Egr-1, PSD-95)	Neuroprotection, Neuronal Function Promotion	Age-related Cognitive Decline	[175]
anti-miR-210	UCMSCs	EVs	-	Immunosuppressive Properties	Psoriasis	[176]
miR-223	UCMSCs, (BMMSCs)	Exosomes	ICAM-1	T cell Adhesion/Migration/Infiltration Inhibition, Inflammatory Factors Suppression	Acute Graft-versus-Host Disease	[177]
anti-miR-301a-3p	UCMSCs	-	IGF-1, PI3K/Akt/FOXO3a	Burn-induced Apoptosis/Organ Vascular Permeability Inhibition	Vascular Endothelial Barrier Dysfunction	[178]
miR-302d-3p	UCMSCs	Exosomes	FLT4, VEGFR3/AKT	Migration/Tube Formation/Lymphangiogenesis Inhibition	IBD	[179]
miR-326	UCMSCs	Exosomes	NEDD8, NF-κB	Neddylation/Inflammation Inhibition	IBD	[180]
miR-335-5p	UCMSCs	Exosomes	ADAM19	Inflammation/EMT Inhibition	Renal Fibrosis	[181]
miR-342-3p	UCMSCs	Exosomes	EDNRA	Thrombus Formation Inhibition, Angiogenesis Promotion	Deep Vein Thrombosis	[182]
miR-378	UCMSCs	EVs	PSMD14/TGF-β1/Smad2/3	Mesangial Hyperplasia/Fibrosis/Proliferation Suppression	Mesangial Proliferative Glomerulonephritis (MsPGN)	[183]
miR-378a-5p	UCMSCs	Exosomes	NLRP3	Macrophage Pyroptosis/Inflammation Inhibition, Cell Survival Promotion	IBD/Colitis	[184]
miR-455-3p	UCMSCs	-	PAK2	Profibrogenic Markers Suppression	Liver Fibrosis	[185]
miR-499	WJMSCs	-	TGFβR 1/3	Creatine Kinase Decrease, Muscle Regeneration, Apoptosis/Fibrosis Inhibition	Duchenne Muscular Dystrophy	[186]
miR-627-5p	UCMSCs	Exosomes	FTO	Cell Survival Promotion, Apoptosis Inhibition, Glucose/Lipid Metabolism Improvement	Non-Alcoholic Fatty Liver Disease	[187]
miR-1246	UCMSCs	Exosomes	PRSS23/Snail/α-SMA	Angiogenesis Promotion, Apoptosis/Hypoxic Damage Reduction	Chronic Heart Failure	[188]
miR-1348a-3p	UCMSCs	Exosomes	Serpine1	Vascular Smooth Muscle Cell Phenotypic Switching/Migration Inhibition	Neointimal Hyperplasia	[189]

UCMSCs: Umbilical Cord-derived Mesenchymal Stem Cells, UCBMSCs: Umbilical Cord Blood-derived Mesenchymal Stem Cells, WJMSCs: Wharton’s Jelly-derived Mesenchymal Stem Cells, EVs: Extracellular vesicles, SLE: Systemic Lupus Erythematosus, DAH: Diffuse Alveolar Hemorrhage, IBD: Inflammatory Bowel Disease.

### 3.3. Regenerative Medicine (Wound Healing/Bone Regeneration)

UCMSCs have additional noteworthy parts to play in the general area of regenerative medicine, especially in bone and skin regeneration, as their unique proliferative and angiogenic biological properties appear to be directly involved in the repair and reconstruction of bone fractures and epithelial injuries. miR-21 is yet again one of the most commonly mentioned microRNAs involved in both cases. Its delivery through UCMSC-EVs was reported to promote corneal wound healing both in vitro and in vivo, via the modulation of the PTEN/PI3K/Akt pathway [70], while miR-21-5p along with miR-125b-5p, transferred through UCBMSC-exosomes, could potentially target and inhibit TGF-β receptors I and II to prevent scar formation and increase wound closure rate [190]. Exosomal miR-150-5p was found to also target the PTEN/PI3K/Akt pathway to facilitate skin wound restoration [108], while miR-125b was further implicated in the promotion of wound healing under hypoxic conditions through the inhibition of TP53INP1-induced endothelial cell apoptosis [191]. EV-transferred miR-27b was shown to target Itchy E3 ubiquitin protein ligase (ITCH) in order to enhance cutaneous wound healing in model mice [83], while miR-17-5p was observed to act similarly in the case of diabetic wounds by regulating the PTEN/AKT/HIF-1α/VEGF axis and ameliorating the impaired angiogenic abilities of endothelial cells in a high glucose environment [64].

In regard to bone repair, exosomal miR-21 injection was proposed as a novel therapeutic intervention for osteonecrosis of the femoral head (ONFH), with miR-21 suppressing the expression of SOX5 and EZH2 [74] as well as regulating the PTEN/Akt pathway [71] to enhance angiogenesis, osteogenesis, and osteocyte survival in injured rats. miR-21-containing exosomes, artificially reinforced with hyaluronic acid hydrogel and nanohydroxyapatite/poly-ε-caprolactone, were also observed to enhance the healing of large bone defects through the NOTCH1/DLL4 pathway and the induction of the angiogenetic process [72]. Additionally, recent studies have reported that several other UCMSC-derived miRNAs have a mediating role to play when it comes to the clinical management of osteoarthritis (OA) and cartilage-related injuries. miR-122-5p, miR-148a-3p, miR-486-5p, miR-let-7a-5p, and miR-100-5p were identified as highly expressed in hUCMSC-EVs, and their presence might explain the chondroprotective properties of hUCMSC-EV treatment, since all five microRNAs were reported to be capable of preventing cartilage degradation by promoting M2 instead of M1 macrophage polarization and easing inflammatory and immune system reactions in OA rats [192]. Treatment with exosomes carrying miR-140-3p was discovered to improve joint injury as well as reduce chondrocyte cell apoptosis in rats with rheumatoid arthritis (RA) through the targeting of SGK1 [101], while intra-articular injection with UCMSCs previously transfected with miR-140-5p could further induce the chondrogenetic and self-healing capacity of injured cartilage tissue in OA rats [102]. miR-181c-5p, delivered by hUCMSC-EVs, was shown to contribute to the repair of cartilage injuries by negatively regulating SMAD7 and promoting BMP2 expression, which in turn amplifies the proliferation, migration, and osteoblastic differentiation capacity of bone marrow stem cells [111].

WJMSC-derived small extracellular vesicles appear to function as natural nanomaterials and through the delivery of multiple microRNAs, including let-7e-5p, miR-423-5p, miR-199a-3p, miR-125b-5p, miR-142-3p, and miR-92a-3p, and the regulation of their various target genes, they were reported to contribute to the maintenance of cell homeostasis in chondrocytes, enhancing proliferation, migration, and infiltration of M2 macrophages while suppressing injury degeneration in in vivo osteoarthritis (OA) model rats [193]. Furthermore, exosomal miR-100-5p and EV-miR-1208 were both demonstrated to inhibit OA development and progression by regulating their target genes NOX4 [93] and METTL3 [125], respectively.

Interestingly, attempts to produce specifically engineered UCMSCs containing gelatin methacrylate and nanoclay hydrogel demonstrated particularly promising results for cartilage defect regeneration through the stable release of EVs that carry miR-23a-3p, which successfully induces PTEN/AKT pathway activity [78]. It should also be noted that UCMSC priming with different cytokines was recently discovered to alter the composition of the miRNA UCMSC-EV cargo, ultimately determining the EVs’ therapeutic effectiveness [109]. As such, UCMSC-EVs previously treated with anti-inflammatory cytokines such as TGF-β and IFN-α presented reduced packaging of miR-181b-3p, which negatively modulates chondrocyte proliferation and regeneration and might naturally contribute to several chondrocyte-associated diseases [109].

### 3.4. Anti-Cancer Treatment

An increasing number of studies suggest that UCMSC-exosomes could also play a mediating role when it comes to the treatment of various cancer types, since many of their cargo miRNAs are reportedly associated with several hallmark oncogenic processes (Table 4). UCMSC-derived exosomes transfected with miR-21-5p were shown to possess tumor-suppressing properties in the case of breast cancer cells by directly targeting ZNF367 and suppressing its migration and invasion properties [194]. Similarly, UCMSC-exosomes modified to overexpress miR-148b-3p were found to inhibit tumor formation and the process of EMT in mice by blocking the oncogenic properties of breast cancer cells and promoting their apoptosis [195], while two more novel anti-oncogenic microRNAs, miR-3182 and miR-3143, were identified as rather promising candidates for the exosome-based therapy of triple negative cancer, with their targets revealed as key genes in cancer pathogenesis [196,197]. On the other hand, miR-224-5p was found highly upregulated in both breast cancer cells and tissues, promoting the autophagic and the oncogenic activity of breast cancer cells by targeting HOXA5 [198]. Interestingly, UCMSC-derived exosomes transfected with the miR-224-5p inhibitor were shown to decrease the proliferation and viability levels of breast cancer cells as well as reduce the volume of tumor tissue in model mice [198].

Delivery of exosomes and EVs loaded with anti-tumorigenic microRNAs also proved efficient in the potential treatment of endometrial and ovarian cancers. miR-503-3p [199] and miR-302a [200] were both found to be capable of suppressing endometrial cancer progression via the inhibition of MEST and cyclin D1, respectively, while miR-200c, combined with UCMSC-secreted IL-21, presented the same attenuating effects in the case of ovarian adenocarcinoma cells through inhibition of the Wnt/β-catenin pathway [201]. Additionally, exosomal microRNA-146a appears to sensitize ovarian cancer cells to the chemotherapeutic agents docetaxel and taxane via the LAMC2/PI3K/Akt axis [202], a property that could prove quite beneficial when it comes to tumor chemoresistance in the clinical setting. High-grade gliomas are another example of highly malignant tumors for which there presently exist no curative treatment protocols and thus for which the need for novel therapeutic methods is imperative. Synthetic miR-124 and miR-145 delivered through MSCs of various origins, including the umbilical cord, were shown to successfully suppress SCP-1 and Sox2, respectively, in order to decrease the migration of cancer cells and the self-renewal abilities of glioma stem cells [203], while WJMSC-delivered miR-124 was suggested to increase the sensitivity of glioblastoma cells to temozolomide treatment [204]. Exosomes carrying packaged lncRNA PTENP1 were also reported to be capable of suppressing the oncogenic activity of miR-10a-5p in glioma cells and enhancing PTEN expression, ultimately decreasing cancer cell viability [205].

Several encouraging studies on the effects of UCMSC-exosome delivery exist in the case of cholangiocarcinoma, as exosomal miR-15a-5p and miR-127-3p were demonstrated to regulate a number of key oncogenic cancer cell properties, including growth, invasion, metastasis, apoptosis, and EMT, through the downregulation of their respective target genes, CHEK1 [206] and ITGA6 [207]. miR-320a was revealed to function in a similar tumor-suppressing manner in lung cancer cells via the SOX4/Wnt/β-catenin pathway [208], while in contrast, EV-derived miR-410 was shown to promote the malignant properties of lung adenocarcinoma cells [209], suggesting that its artificial inhibition could perhaps pose a future therapeutic option. The beneficial effects of UCMSC-transferred microRNAs were also explored in nasopharyngeal and esophageal cancer, with exosome-delivered miR-181a significantly delaying nasopharyngeal carcinoma progression via the inhibition of KDM5C [210], while microRNAs miR-655-3p and microRNA-375 downregulated HIF-1α [211] and ENAH [212], respectively, to achieve the same suppressing effect in esophageal carcinoma. Additionally, quite recently the anti-tumor properties of exosome and EV-enclosed microRNAs were proven to be efficient in the cases of colorectal [213], gastric [214], bladder [215], thyroid [216], and prostate [217] cancer, as well as in hepatocellular carcinoma [218], pancreatic ductal adenocarcinoma [219], Wilms tumor [220], and chronic myeloid leukemia [221], further affirming that UCMSC-transferred microRNAs could indeed become valuable tools for prospective clinical applications in the field of cancer therapy.

**Table 4 ijms-24-09189-t004:** miRNA-guided applications of perinatal and neonatal MSCs in anti-cancer treatment.

miRNA	Tissue Origin	Vehicle Type	Target (Gene/Pathway)	Function	Clinical Application	Reference
miR-10a-5p	UCMSCs	Exosomes	PTEN	Cell Growth/Cell Survival Promotion	Glioma	[205]
miR-15a-5p	UCMSCs	Exosomes	CHEK1	Proliferation/Invasion/Migration/EMT Inhibition, Apoptosis Promotion	Cholangiocarcinoma	[206]
	UCMSCs	Exosomes	SEPT2	Proliferation/Migration/Invasion Inhibition, Apoptosis Promotion	Wilms Tumor	[220]
miR-21-5p	UCMSCs	Exosomes	ZNF367	Migration/Invasion Inhibition	Breast Cancer	[194]
miR-30c-5p	UCMSCs	EVs	PELI1/PI3K/AKT	Proliferation/Migration Inhibition	Papillary Thyroid Carcinoma	[216]
miR-100-5p	UCMSCs	Exosomes	-	Proliferation/Migration Promotion	Pancreatic Ductal Adenocarcinoma	[222]
miR-124	WJMSCs	-	CDK6	Proliferation/Migration Suppression, Chemosensitivity Promotion	Glioblastoma Multiform	[204]
miR-125b	WJMSCs	EVs	HIF1α	Proliferation/EMT/Angiogenesis Inhibition	Triple Negative Breast Cancer	[223]
miR-127-3p	UCMSCs	EVs	ITGA6/TGF-β1/Smad	Proliferation/Invasion/Migration/EMT Inhibition, Apoptosis Promotion	Cholangiocarcinoma	[207]
miR-139-5p	UCMSCs	Exosomes	PRC1	Proliferation/Migration/Invasion Suppression	Bladder Cancer	[215]
miR-145-5p	UCMSCs	Exosomes	TGF-β/Smad3	Proliferation/Invasion Inhibition, Apoptosis/Cell Cycle Arrest Promotion	Pancreatic Ductal Adenocarcinoma	[219]
miR-145a-5p	UCMSCs	Exosomes	USP6/GLS1	Apoptosis/Chemosensitivity Promotion	Chronic Myeloid Leukemia	[221]
miR-146a	UCMSCs	Exosomes	LAMC2/PI3K/Akt	Cell Growth Suppression, Chemosensitivity Promotion	Ovarian Cancer	[202]
miR-148b-3p	UCMSCs	Exosomes	TRIM59	Proliferation/Invasion/Migration Inhibition, Apoptosis Promotion	Breast Cancer	[195]
miR-181a	UCMSCs	Exosomes	KDM5C	Cell Growth Suppression	Nasopharyngeal Carcinoma	[210]
miR-200c	UCMSCs	-	Wnt/β-catenin	Tumor Growth/Metastasis Inhibition	Ovarian Cancer	[201]
anti-miR-224-5p	UCMSCs	Exosomes	HOXA5	Proliferation Inhibition, Apoptosis Promotion	Breast Cancer	[198]
miR-302a	UCMSCs	EVs	cyclin D1/AKT	Proliferation/Migration Inhibition	Endometrial Cancer	[200]
miR-320a	UCMSCs	Exosomes	SOX4/Wnt/β-catenin	Proliferation/Metastasis Inhibition	Lung Cancer	[208]
miR-375	UCMSCs	Exosomes	ENAH	Proliferation/Invasion/Migration/Tumorsphere Formation Inhibition, Apoptosis Promotion	Esophageal Squamous Cell Carcinoma	[212]
anti-miR-375	UCMSCs	Exosomes	PTPN4/STAT3	Proliferation/Migration/Invasion/Chemoresistance Suppression, Apoptosis Promotion	Prostate Cancer	[217]
miR-410	UCMSCs	EVs	PTEN	Proliferation Promotion, Apoptosis Suppression	Lung Adenocarcinoma	[209]
miR-431-5p	UCMSCs	Exosomes	PRDX1	Cell Growth Suppression, Prognostic Marker	Colorectal Cancer	[213]
miR-451a	UCMSCs	Exosomes	ADAM10	Paclitaxel Resistance/Cell Cycle Transition/Proliferation/Migration/Invasion Inhibition, Apoptosis Promotion	Hepatocellular Carcinoma	[218]
miR-503-3p	UCBMSCs	Exosomes	MEST	Cell Growth Suppression	Endometrial Cancer	[199]
miR-655-3p	UCMSCs	EVs	HIF-1α/LMO4/HDAC2	Cell Growth/Metastasis Suppression	Esophageal Cancer	[211]
miR-3182	UCMSCs	Exosomes	mTOR, S6KB1	Proliferation/Migration Inhibition, Apoptosis Promotion	Triple Negative Breast Cancer	[197]
miR-6785-5p	UCMSCs	Exosomes	INHBA	Angiogenesis/Metastasis Suppression	Gastric Cancer	[214]
miR-11401	UCMSCs	EVs	SCOTIN/p53	Doxorubicin-induced Apoptosis Inhibition	Chemotherapy	[224]

UCMSCs: Umbilical Cord-derived Mesenchymal Stem Cells, UCBMSCs: Umbilical Cord Blood-derived Mesenchymal Stem Cells, WJMSCs: Wharton’s Jelly-derived Mesenchymal Stem Cells, EVs: Extracellular vesicles, EMT: Epithelial-to-Mesenchymal Transition.

Finally, apart from separate miRNAs, miRNA groups and signatures are being exploited for their ability to guide UCMSC-based regenerative and therapeutic applications (Table 5).

## 4. Future Directions

As thoroughly described, the exploitation of miRNA regulatory properties along with the unique characteristics of UCMSCs is a quite promising research field with a multitude of applications, expanding from aesthetic interventions to therapeutic regimes. Nevertheless, there are currently certain limitations that need to be overcome for the successful implementation of miRNA-based therapies in the clinic. Upon administration, the first obstacle that naked and unmodified miRNA molecules need to overcome is low cell membrane penetration [235], followed by rapid degradation and clearance from the circulation [236,237]. A solution to these problems is the use of evolved delivery vehicles, including nanocarriers and exosomes, that stabilize and protect miRNAs until they reach their targets. Still, targeting of extra-hepatic tissues remains a barrier, thus limiting the use of miRNA-based therapies. Although lipid and polymer nanoparticles can achieve efficient cellular uptake and stability of delivered miRNAs, they selectively accumulate in the liver instead of the intended target tissues [238]. Of high significance are also the off-target effects of miRNAs, which are mainly attributed to the multiple intracellular targets of each miRNA [239], along with their ability to abnormally activate the host’s immune system [240]; These effects can lead to serious toxicities and adverse events that lower the therapeutic value of miRNA-based therapies. The combined use of MSCs could alleviate these effects, given the immunosuppressive effects of the latter.

Currently, there are a few clinical studies in different stages of processing (https://clinicaltrials.gov). In some of those, MSCs from different sources are evaluated for their safety and efficacy against metabolic and autoimmune diseases based on the expression levels of certain miRNAs among other biomarkers. Accordingly, the intracorporal delivery of MSC-derived exosomes considered to carry several essential elements, including miRNAs, is the main subject of investigation in ongoing phase I/II studies. One completed phase I trial that examined the biodistribution and safety of inhaled MSC-EVs in healthy volunteers revealed no adverse events after seven days of treatment (PMID: 34429860). Interestingly, there are two registered clinical trials assessing the efficacy of MSC exosome inhalation in coronavirus disease 2019 (COVID-19) patients; the completed one determined that inhalation of exosomes twice a day for a total period of 10 days increased SpO2 concentration in the blood of patients with pneumonia in the absence of any adverse events (https://clinicaltrials.gov/NCT04491240 (accessed on 20 May 2023)).

Overall, miRNA-driven multi-lineage differentiation of UCMSCs forms a subcategory of the field of MSC-based therapeutics. Recent advances in the field, combined with the putative cell-free implementation of UCMSC-EVs carrying key miRNAs, pave the way for their successful application in therapeutic and regenerative remedies that will enhance cure rates in the absence of serious side effects. Importantly, the evolution of induced pluripotent stem cell (iPSC) research provides the ability to obtain large numbers of iPSC-derived MSC-like cells with properties similar to their native counterparts, thus offering a double advantage: an abundance of cells that can be exploited in therapeutic protocols with a concomitant absence of ethical issues. At the same time, since altered miRNA levels are detected in cells derived from differentiated MSCs, these miRNAs can be utilized as powerful biomarkers of the differentiation process and the quality of the descendant cells, as well.

## Figures and Tables

**Figure 1 ijms-24-09189-f001:**
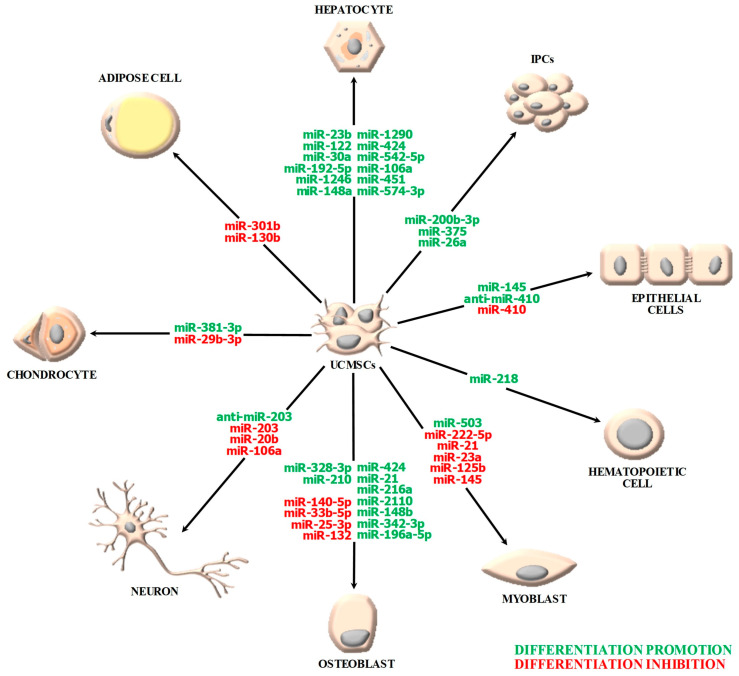
The implication of distinct microRNAs (miRNAs) in the differentiation process of umbilical cord-derived mesenchymal stem cells (UCMSCs). MiRNAs that promote the differentiation of UCMSCs are shown in green, while those that inhibit this process are designated in red. IPCs: Insulin-Producing Cells.

**Figure 2 ijms-24-09189-f002:**
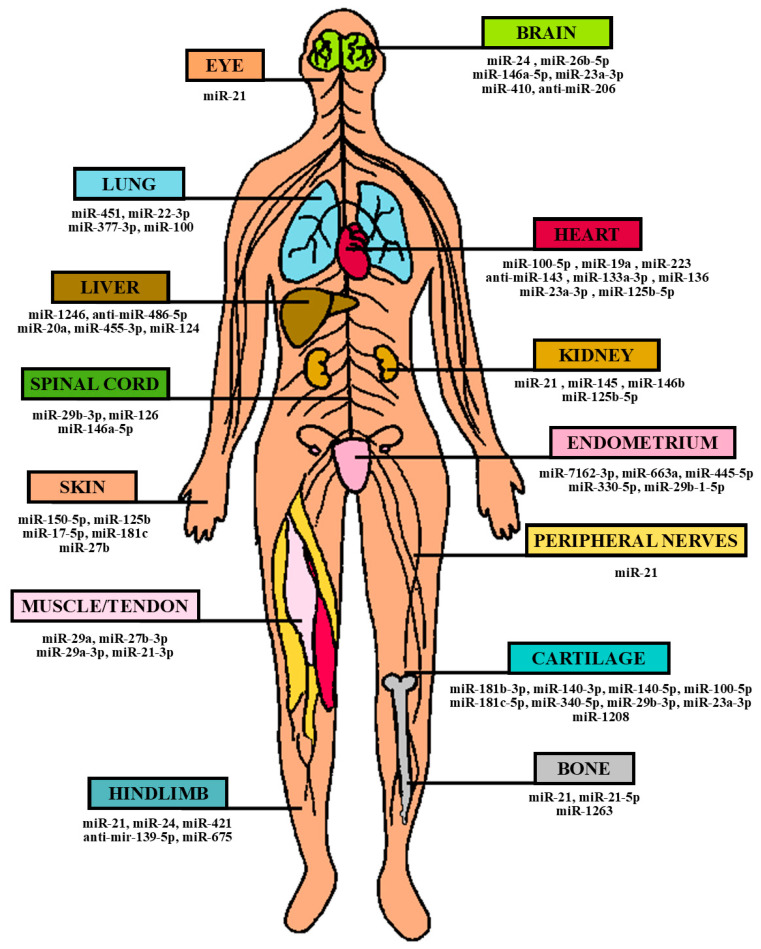
Clinical applications of umbilical cord-derived stem cells in the context of microRNA (miRNA)-directed regenerative/healing potential.

**Figure 3 ijms-24-09189-f003:**
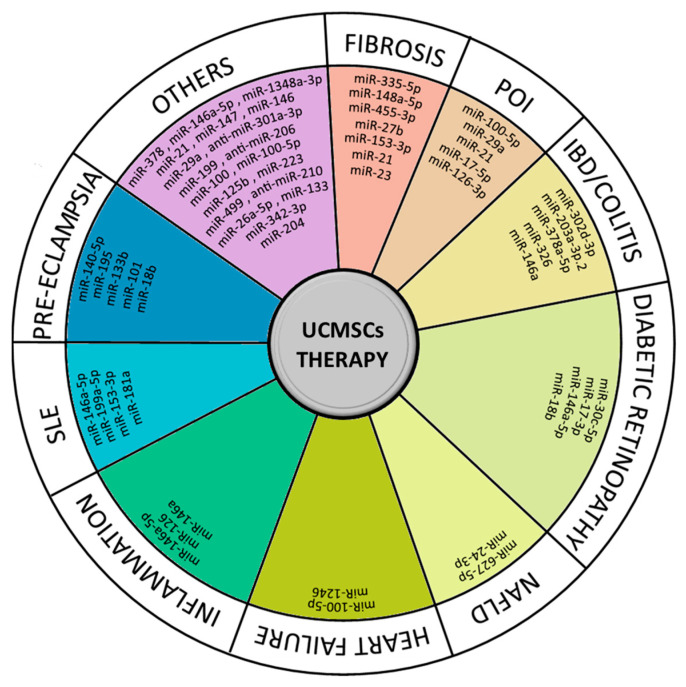
Summary of microRNA (miRNA)-guided therapeutic applications of umbilical cord-derived mesenchymal stem cells (UCMSCs). SLE: Systemic Lupus Erythematosus; POI: Primary Ovarian Insufficiency; IBD: Inflammatory Bowel Disease; NAFLD: Non-alcoholic Fatty Liver Disease.

**Table 1 ijms-24-09189-t001:** Detailed roles of distinct microRNAs (miRNAs) in the differentiation process of mesenchymal stem cells (MSCs).

miRNA	Tissue Origin	Target (Gene/Pathway)	Target Regulation	miRNA Function	Reference
miR-424	WJMSCs	BMP2	Up	Osteogenic	[26]
miR-140-5p	UCMSCs, ATMSCs, BMMSCs	BMP2	Down	Anti-Osteogenic	[27]
miR-342-3p	UCMSCs	Sufu/Shh	Down/Up	Osteogenic	[28]
miR-196a-5p	WJMSCs	SERPINB2	Down	Osteogenic	[29]
miR-21	UCMSCs	KLF12/Wnt/β-catenin	Down/Up	Osteogenic	[30]
miR-21	UCMSCs	PI3K/β-catenin	Up	Osteogenic	[31]
miR-2110	UCMSCs (Exosomes)	(TNF)	(Up)	Osteogenic	[32]
miR-328-3p	UCMSCs (Exosomes)	CHRD	Down	Osteogenic	[32]
miR-25-3p	UCMSCs	SMAD5/(Dlx5/OSX)	Down/(Down)	Anti-Osteogenic	[33]
miR-33b-5p	UCMSCs	Wnt10b/(Dlx5/OSX)	Down/(Down)	Anti-Osteogenic	[33]
miR-132	UCMSCs	Osterix/Wnt/β-catenin	Down/Down	Anti-Osteogenic	[34]
miR-216a	UCMSCs, ATMSCs	c-Cbl/PI3K/AKT	Down/Up	Osteogenic	[35]
miR-122-5p	UCMSCs	SOX11, VIM, (*CUTL1*, *CAT1*)	Down	Hepatogenic	[36]
miR-106a, miR-574-3p, miR-451	UCMSCs	Sox17, FoxA2, AFP	Up	Hepatogenic	[37]
miR-20b, miR-106a	WJMSCs	Ngn2, MAP2, TUBB3	Down	Anti-Neurogenic	[38]
miR-203	UCBMSCs	DKK1, CRX, NRL	Down	Anti-Neurogenic	[39]
miR-29b-3p	UCMSCs	SOX9	Down	Anti-Chondrogenic	[40]
miR-381	UCMSCs (sEVs)	SOX9/TAOK1	Up/Down	Chondrogenic	[41]
miR-410	UCBMSCs	MITF, LRAT, RPE65, Bestrophin, EMMPRIN	Down	Anti-Epithelial	[42]
miR-145	UCMSCs	TGFβRII	Down	Epithelial	[43]
miR-200b-3p	UCMSCs	ZEB2	Down	IPCs Differentiation	[44]
miR-375, miR-26a	N-UCMSCs (chicken)	MTPN, SOX6, BHLHE22, CCND1	Down	IPCs Differentiation	[45]
miR-218	UCMSCs	MITF, (HoxB4, NF-Ya)	Down, (Up)	Hematopoietic	[46]
miR-301b/miR-130b	UCMSCs, ATMSCs, BMMSCs	PPARγ	Down	Anti-Adipogenic	[47]
miR-21, miR-23a, miR-125b, miR-145	UCMSCs (Exosomes)	TGF-β2, TGF-βR2, SMAD2	Down	Anti-Myofibroblastic	[48]
miR-503-5p	UCMSCs	SMAD7	Down	Myogenetic	[49]
miR-222-5p	UCMSCs	ROCK2, αSMA	Down	Anti-Myogenetic	[49]

WJMSCs: Wharton’s Jelly-derived Mesenchymal Stem Cells, UCMSCs: Umbilical Cord-derived Mesenchymal Stem Cells, UCBMSCs: Umbilical Cord Blood-derived Mesenchymal Stem Cells, ATMSCs: Adipose Tissue-derived Mesenchymal Stem Cells, BMMSCs: Bone Marrow-derived Mesenchymal Stem Cells, N-UCMSCs: Nestin-positive Umbilical Cord-derived Mesenchymal Stem Cells, sEVs: Small Extracellular Vesicles, IPC: Insulin-Producing Cells.

**Table 2 ijms-24-09189-t002:** miRNA-guided applications of perinatal and neonatal MSCs in regenerative medicine.

miRNA	Tissue Origin	Vehicle Type	Target (Gene/Pathway)	Function	Clinical Application	Reference
miR-17-5p	UCMSCs	EVs	PTEN/AKT/HIF-1α/VEGF	Angiogenesis/Proliferation/Migration/ Tube Formation Promotion	Diabetic Wound Healing	[64]
miR-19a	UCMSCs	Exosomes	SOX6, AKT/JNK3/caspase-3	Hypoxic Damage Reduction	AMI	[65]
miR-20a	UCMSCs	Exosomes	Beclin-I, FAS, Caspase-3, mTOR, P62, LC3II	Apoptosis/Autophagy Inhibition	Hepatic IRI	[66]
miR-21	UCMSCs	MVs	PDCD4	Apoptosis Inhibition	Renal IRI	[67]
	UCBMSCs	-	CHIP/HIF-1α	Proliferation/Migration/Angiogenesis/ Neovascularization Promotion	CLI	[68]
	UCMSCs	EVs	PI3K/AKT	Proliferation Promotion	Peripheral Nerve Injury	[69]
	UCMSCs	EVs	PTEN/PI3K/Akt	Proliferation/Migration Promotion	Corneal Wound Healing	[70]
	WJMSCs	Exosomes	PTEN/Akt	Angiogenesis/Osteogenesis/Cell Survival Promotion	ONFH	[71]
	UCMSCs	Exosomes	NOTCH1/DLL4	Angiogenesis Promotion	Bone Regeneration	[72]
miR-21-3p	UCMSCs	Exosomes	p65, COX2	Fibrosis/Inflammation Inhibition	Tendon Injury	[73]
miR-21-5p	UCMSCs	Exosomes	SOX5, EZH2	Angiogenesis/Osteogenesis Promotion	ONFH	[74]
miR-22-3p	UCBMSCs	Exosomes	FZD6	Inflammation/Oxidative Stress/Apoptosis Suppression, Proliferation Promotion	Acute Lung Injury	[75]
miR-23a-3p	UCBMSCs	Exosomes	-	M2 Macrophage Polarization Promotion, Microglia Activation Inhibition	Cerebral Infarction	[76]
	UCBMSCs	Exosomes	DMT1	Ferroptosis Inhibition	AMI	[77]
	UCMSCs	EVs	PTEN/AKT	Differentiation/Proliferation/Migration Promotion	Cartilage Regeneration	[78]
miR-24	UCMSCs	EVs	AQP4/P38 MAPK/ERK1/2/P13K/AKT	Apoptosis/Inflammation Inhibition, Proliferation/Migration Promotion	Cerebral IRI	[79]
	UCMSCs	Exosomes	Bim	Immune Rejection Prevention	Ischemic Hindlimb Injury	[80]
miR-26b-5p	UCMSCs	Exosomes	CH25H-/TLR	Microglia M1 Polarization Inhibition	Cerebral IRI	[81]
	UCMSCs	Exosomes	MAT2A/MAPK/STAT3	Apoptosis/Inflammatory Inhibition	EBI	[82]
miR-27b	UCMSCs	EVs	ITCH/JUNB/IRE1α	Proliferation/Migration Promotion	Skin Wound Healing	[83]
miR-27b-3p	UCMSCs	Exosomes	ARHGAP5/RhoA	Proliferation/Invasion Promotion	Tendon Injury	[84]
miR-29a	WJMSCs	-	TIMP-2	Angiogenesis, Blood Perfusion Promotion	Skeletal Muscle Ischemic Injury	[85]
miR-29a-3p	UCMSCs	Exosomes	PTEN/mTOR/TGF-β1	Differentiation/Healing Promotion	Tendon Injury	[86]
miR-29b-3p	UCMSCs	EVs	PTEN/Akt/mTOR	Necrosis Reduction, Nerve Function Promotion	SCI	[87]
	UCMSCs	EVs	PTEN/PI3K/AKT	Neuronal Apoptosis Inhibition	SCI	[88]
	UCMSCs	Exosomes	FoxO3	Apoptosis/Senescence Promotion, Migration/Inflammation Inhibition	Cartilage Defect	[89]
miR-29b-1-5p	WJMSCs	-	RAP1B/VEGF	Angiogenesis Promotion/Tissue Repair	Damaged Endometrium	[90]
miR-100	WJMSCs	MVs	mTOR	Autophagy Promotion	Acute Lung Injury	[91]
miR-100-5p	UCMSCs	Exosomes	FOXO3, NLRP	Ιnflammasome/Cytokine Activation Suppression, Pyroptosis Inhibition	AMI	[92]
	UCMSCs	Exosomes	NOX4	ROS Production/Apoptosis Inhibition	Osteoarthritis	[93]
miR-124	UCBMSCs	Exosomes	Foxg1	Proliferation/Regeneration Promotion, Injury Inhibition	Liver Regeneration	[94]
miR-125b	UCMSCs	Exosomes	TP53INP1	Cell Survival/Migration Promotion, Apoptosis Inhibition	Skin Wound Healing	[75]
miR-125b-5p	UCMSCs	Exosomes	SMAD7	Hypoxic Injury/Apoptosis Promotion	AMI	[95]
	UCMSCs	Exosomes	p53	Cell cycle arrest/Apoptosis Inhibition	Ischemic AKI	[96]
miR-126	UCMSCs	Exosomes	SPRED1, PIK3R2	Neurogenesis/Angiogenesis Promotion, Apoptosis Inhibition	SCI	[97]
miR-133a-3p	UCMSCs	MIF-Exosomes	AKT/VEGF	Angiogenesis/Proliferation Promotion, Apoptosis/Fibrosis Inhibition	AMI	[98]
miR-136	UCMSCs	Exosomes	Apaf1	Apoptosis/Fibrosis/Senescence Suppression, Angiogenesis Promotion	Age-associated MI	[99]
anti-miR-139-5p	UCBMSCs/UCMSCs	-	-	Proliferation/Migration/Angiogenesis Promotion	Peripheral Arterial Disease (Limb Ischemia)	[100]
miR-140-3p	UCMSCs	EVs	SGK1	Inflammation/Oxidative Stress/Fibrosis Suppression	Rheumatoid Arthritis	[101]
miR-140-5p	UCMSCs	-	-	Chondrogenesis/Cartilage Self-Repair Promotion	Osteoarthritis	[102]
anti-miR-143	UCMSCs	Exosomes	Bcl-2/Beclin-1	Apoptosis/Autophagy Inhibition	Cardiac IRI	[103]
miR-145	UCMSCs	-	PI3K/AKT/mTOR	Autophagy Promotion	AKI	[104]
miR-146a-5p	UCMSCs	Exosomes	IRAK1/TRAF6	Neuroinflammation Suppression	Ischemic Stroke	[105]
	UCMSCs	Exosomes	Traf6/Irak1/NFκB	Neurotoxic Astrocytes Reduction	SCI	[106]
miR-146b	UCMSCs	-	IRAK1/NF-κB	Apoptosis/Inflammation Inhibition	Sepsis-associated AKI	[107]
miR-150-5p	UCMSCs	Exosomes	PTEN/PI3K/Akt	Growth/Migration Promotion, Apoptosis Inhibition	Skin Wound Healing	[108]
miR-181b-3p	UCMSCs	EVs	-	Chondrocyte Proliferation/Cartilage Regeneration Inhibition	Osteoarthritis	[109]
miR-181c	WJMSCs	Exosomes	TLR4/NF-κB/p65	Burn-induced Inflammation/Macrophage Inflammation Suppression	Burn Injury	[110]
miR-181c-5p	UCMSCs	EVs	SMAD7	Proliferation/Migration/Chondrogenesis Promotion	Cartilage Injury	[111]
anti-miR-206	UCMSCs	Exosomes	BDNF/TrkB/CREB	Apoptosis Inhibition	EBI	[112]
miR-223	UCMSCs	EVs	P53/S100A9	Angiogenesis Promotion, Inflammation/Fibrosis Suppression	MI	[113]
miR-330-5p	UCMSCs	SF-SIS Scaffold	CircPTP4A2, PDK2	Mitochondrial Metabolism Impairment, Fibrosis Promotion	Endometrial Hypoxic Injury	[114]
miR-340-5p	WJMSCs	Exosomes	IL4	Osteochondral Regeneration	Articular Cartilage Defect	[115]
miR-377-3p	UCMSCs	Exosomes	RPTOR	Autophagy Promotion	Acute Lung Injury	[116]
miR-410	UCMSCs	EVs	HDAC1/EGR2/Bcl2	Apoptosis Inhibition	HIBD	[117]
miR-421	UCMSCs	Exosomes	FOXO3a	Pyroptosis/Ischemic Injury Promotion	Acute Lower Limb Ischemic Injury	[118]
miR-451	UCMSCs	Exosomes	TLR4/NF-κB, MIF/PI3K/Akt	Inflammation Suppression, M2 Macrophage Polarization Activation	Acute Lung Injury	[119]
miR-455-3p	UCMSCs	Exosomes	PIK3r1/IL-6	Macrophages Activation Inhibition, Inflammatory Cytokine Suppression	Acute Liver Injury	[120]
miR-455-5p	UCMSCs	-	SOCS3/JAK/STAT3	Cell Cycle/Proliferation Promotion, Endometrial Glands Increase, Fibrosis Suppression	Endometrial Injury	[121]
anti-miR-486-5p	UCBMSCs	-	PIM1/TGF-β/Smad	Apoptosis/Inflammation Suppression, Proliferation Promotion	Oxidative Stress Injury	[122]
miR-663a	UCMSCs	Exosomes	CDKN2A	Proliferation Promotion, Apoptosis/Migration/EMT Inhibition	Endometrial Hypoxic Injury	[123]
miR-675	UCMSCs	Exosomes	p21, TGF-β1	Blood Perfusion Promotion, Senescence Inhibition	Ischemia-induced Vascular Dysfunction	[124]
miR-1208	UCMSCs	EVs	METTLE3	Proliferation/Migration Promotion, Apoptosis/Inflammation Inhibition	Osteoarthritis	[125]
miR-1246	UCBMSCs	Exosomes	GSK3β-Wnt/β-catenin	Apoptosis Inhibition	Hepatic IRI	[126]
	UCMSCs	Exosomes	IL-6/gp130/STAT3	Th17/Treg Balance Regulation	Hepatic IRI	[127]
miR-1263	UCMSCs	Exosomes	Mob1/Hippo	Apoptosis Inhibition	Disuse Osteoporosis	[128]
miR-7162-3p	UCMSCs	Exosomes	APOL6	Apoptosis Inhibition, Cell Repair/Regeneration	Endometrial Injury	[129]

UCMSCs: Umbilical Cord-derived Mesenchymal Stem Cells, UCBMSCs: Umbilical Cord Blood-derived Mesenchymal Stem Cells, WJMSCs: Wharton’s Jelly-derived Mesenchymal Stem Cells, EVs: Extracellular vesicles, MVs: Microvesicles, MIF: Macrophage Migration Inhibitor, SF: Silk Fibroin, SIS, Small Intestinal Submucosa, AMI: Acute Myocardial Infarction, IRI: Ischemic/Reperfusion Injury, CLI: Critical Limb Ischemia, ONFH: Osteonecrosis of the Femoral Head, EBI: Early Brain Injury, SCI: Spinal Cord Injury, AKI: Acute Kidney Injury, HIBD: Hypoxia-Ischemia Brain Damage.

**Table 5 ijms-24-09189-t005:** Multiple miRNAs/miRNA signatures that mediate MSC-based regenerative/healing applications.

miRNAs	Tissue Origin	Vehicle Type	Target (Gene/Pathway)	Function	Clinical Application	Reference
miR-199a-3p/145-5p	UCMSCs	Exosomes	Cblb, Cbl, NGF/TrkA	Neuronal Differentiation Promotion	SCI	[131]
miR-let-7a, miR-let-7e	WJMSCs	EVs	Casp3	Apoptosis Inhibition	Hypoxic-ischemic Brain Injury	[225]
miR-21-5p, miR-125b-5p	UCBMSCs	Exosomes	TGF-βI, TGF-βII	Wound Closure Acceleration, Scar Formation Inhibition, Skin/Nerve/Vessel Regeneration	Skin Wound Healing	[190]
miR-21, miR-29, miR-221, let-7a	WJMSCs	EVs	BMP, PI3K/AKT	Osteogenic Differentiation/Osteoblast Activity Promotion	Osteoporosis	[226]
miR-122-5p, miR-148a-3p, miR-486-5p, miR-let-7a-5p, miR-100-5p	UCMSCs	EVs	PI3K/Akt	Cartilage Degradation/Inflammation Suppression, M2 Macrophage Polarization Promotion	Osteoarthritis	[192]
miR-92b-3p, miR-32-5p, let-7b-5p, miR-19a-3p, miR-19b-3p	WJMSCs	Exosomes	DKK3	Osteochondral Regeneration	Articular Cartilage Defect	[115]
miR-23a-3p, miR-221-3p, miR-23b-3p, miR-141-3p, miR-144-3p, miR-200a-3p, miR-454-3p, miR-23c, miR-320b	WJMSCs	Exosomes	CXCL12	Osteochondral Regeneration	Articular Cartilage Defect	[115]
miR-374a-5p, miR-495-3p, miR-323a-3p	WJMSCs	Exosomes	CCL2	Osteochondral Regeneration	Articular Cartilage Defect	[115]
miR-19b-3p, miR-185-5p	WJMSCs	Exosomes	POSTN	Osteochondral Regeneration	Articular Cartilage Defect	[115]
let-7e-5p, miR-423-5p, miR-199a-3p, miR-125b-5p, miR-142-3p, miR-92a-3p	WJMSCs	EVs	ECM-receptor interaction, NOTCH, (*STAT3*, *IGF1R*)	Proliferation/Migration/M2 Infiltration Promotion, Homeostasis Maintenance, Injury Degradation Suppression	Osteoarthritis	[193]
miR-100, miR-146a, (miR-21, miR-221, miR-143)	UCMSCs	Exosomes	-	Cell Cycle/Proliferation Promotion, Apoptosis Inhibition	Vaginal Reconstruction	[227]
miR-135b-5p, miR-499a-3p	UCMSCs	Exosomes	MEF2C	Angiogenesis Promotion	Tissue Regeneration	[228]
miR-136, miR-494, miR-495	UCMSCs	Exosomes	-	Biomarkers (Diagnosis, Evaluation)	Pre-Eclampsia	[229]
miR-21, miR-146a-5p	UCMSCs	Exosomes	PI3K/mTOR	Fertility Recovery, Oocyte Production Promotion	Premature Ovarian Insufficiency	[230]
miR-146a-5p, miR-548e-5p	UCMSCs	Exosomes	NF-κB, AKT, MAPK	Inflammation Suppression, Proliferation/Migration Promotion	Inflammatory Diseases/Preterm Birth	[231]
miR-17 Superfamily	WJMSCs, BMMSCs	Exosomes	(BMPR2)	Proliferation Promotion	Pulmonary Hypertension	[174]
miR-10a-5p, miR-146a-5	UCMSCs	Exosomes	VEGF-VEGFR2/AKT, MEK/ERK	Apoptosis Inhibition	Chronic Obstructive Pulmonary Disease	[232]
miR-24, miR-199a-5p	WJMSCs	Exosomes	-	Muscle Regeneration	Duchenne Muscular Dystrophy	[233]
miR-124, miR-145	Bone Marrow/Adipose Tissue/Placenta/Umbilical Cord MSCs	-	SCP-1, Sox2	Migration/Cell Self-Renewal Inhibition	Glioma	[203]
let-7a-2-3p, let-7d, let-7e, let-7f, let-7f-1-3p, let-7g, let-7i, let-7i-3p, miR-100, miR-106a, miR-106b, miR-125a-3p, miR-125a-5p, miR-126, miR-146a, miR-17, miR-181a, miR-18b, miR-196a-3p, miR-19a, miR-19b, miR-200b, miR-20a, miR-20b, miR-21, miR-210, miR-25, miR-27a-5p, miR-29b, miR-302a, miR-302b-5p, miR-302c-5p, miR-30c, miR-31, miR-32, miR-335, miR-34a, miR-374a, miR-378 miR-3915, miR-3924, miR-601, miR-622, miR-920, miR-92a, miR-93, miR-98	WJMSCs	-	-	Wound Healing Promotion	Excisional/Diabetic Wounds	[234]

UCMSCs: Umbilical Cord-derived Mesenchymal Stem Cells, UCBMSCs: Umbilical Cord Blood-derived Mesenchymal Stem Cells, WJMSCs: Wharton’s Jelly-derived Mesenchymal Stem Cells, MSCs: Mesenchymal Stem Cells EVs: Extracellular Vesicles SCI: Spinal Cord Injury.

## Data Availability

Not applicable.

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
