# Peer review of "miRNA-Guided Regulation of Mesenchymal Stem Cells Derived from the Umbilical Cord: Paving the Way for Stem-Cell Based Regeneration and Therapy"

_ijms, 2023, doi:10.3390/ijms24119189_

Round 1
Reviewer 1 Report
The authors carried out an exhaustive review of microRNAs involved in MSCs differentiation and their use in regenerative medicine mainly carried out in extracellular vesicles.
Very good images and tables.
Ongoing clinical trials using miRNAs have been described.
I would suggest including any current limitations in the use of miRNA therapies in humans or issues that still need to be overcome (in future directions).
Reviewer 2 Report
The work by Thomaidou et al. titled "miRNA-guided regulation of mesenchymal stem cells derived from the umbilical cord: paving the way for stem-cell based regeneration and therapy" is a well written review. The manuscript highlights the field very thoroughly and meticulously explains the various applications while listing the appropriate miRNAs. Appropriate literature is cited and the figures and tables are useful for the understanding of the review. Overall a very nice piece of work. Only suggestion would be to add more about how miRNAs work in general to the introduction section. This is not really covered at all and would be helpful to orient the reader.
Very well written review with minor style issues.
Reviewer 3 Report
The review of A. C. Thomaidou “miRNA-guided regulation of mesenchymal stem cells derived 2 from the umbilical cord…” focuses on various aspects of the effects of microRNA (miRNA) on the cellular physiology of mesenchymal stem cells, with an emphasis on umbilical cord MSCs, the closest to embryonic cells.
The inhibitory and stimulatory effects of miRNAs on different types of MSC differentiation are described, as well as the molecular mechanisms and signaling pathways mediating the effects of individual miRNAs on the differentiation activity of MSCs.
The issues related to clinical application of MSCs mediated by their produced miRNAs (mainly in exosomes and extracellular vesicles) for treatment of acute organ damage and correction of pathological conditions, as well as for tissue healing and regeneration are considered separately.
Overall, the review characterized the wide range of regulatory capabilities and therapeutic potential of the extensive miRNA family.
The work makes a very good impression. The reviewer has no critical comments.

Overall, English is good. Some misprints mentioned by reviewer:
some distorted abbreviations, please see lines 389, 403, 437, 466
Reviewer 4 Report
The authors present here the role of miRNAs in the differentiation of MSCs to various lineages. The article is well written and extensive review of literature has been presented. Following are couple of suggestions for improving the manuscript,
1. The table 5 must be included as a separate section instead of future directions as it just discuses the studies which have shown the relevance of a group of miRNAs for differentiation of MSCs.
2. Future directions section should be revised to discuss the potential future areas of research related to miRNAs and MSCs.
English language quality is good overall.
